# SnSe Nanosheets: From Facile Synthesis to Applications in Broadband Photodetections

**DOI:** 10.3390/nano11010049

**Published:** 2020-12-27

**Authors:** Xiangyang Li, Zongpeng Song, Huancheng Zhao, Wenfei Zhang, Zhenhua Sun, Huawei Liang, Haiou Zhu, Jihong Pei, Ling Li, Shuangchen Ruan

**Affiliations:** 1Shenzhen Key Laboratory of Laser Engineering, College of Physics and Optoelectronic Engineering, Shenzhen University, Shenzhen 518060, China; 2170285209@email.szu.edu.cn (X.L.); 1800281011@email.szu.edu.cn (H.Z.); zhangwf@szu.edu.cn (W.Z.); szh@szu.edu.cn (Z.S.); hwliang@szu.edu.cn (H.L.); 2College of New Materials and New Energies, Shenzhen Technology University, Shenzhen 518118, China; songzongpeng1986@icloud.com (Z.S.); zhuhaiou@sztu.edu.cn (H.Z.); 3College of Electronics and Information Engineering, Shenzhen University, Shenzhen 518060, China; jhpei@szu.edu.cn

**Keywords:** probe sonication, carrier dynamics, SnSe NSs/graphene, photodetectors

## Abstract

In recent years, using two-dimensional (2D) materials to realize broadband photodetection has become a promising area in optoelectronic devices. Here, we successfully synthesized SnSe nanosheets (NSs) by a facile tip ultra-sonication method in water-ethanol solvent which was eco-friendly. The carrier dynamics of SnSe NSs was systematically investigated via a femtosecond transient absorption spectroscopy in the visible wavelength regime and three decay components were clarified with delay time of τ_1_ = 0.77 ps, τ_2_ = 8.3 ps, and τ_3_ = 316.5 ps, respectively, indicating their potential applications in ultrafast optics and optoelectronics. As a proof-of-concept, the photodetectors, which integrated SnSe NSs with monolayer graphene, show high photoresponsivities and excellent response speeds for different incident lasers. The maximum photo-responsivities for 405, 532, and 785 nm were 1.75 × 10^4^ A/W, 4.63 × 10^3^ A/W, and 1.52 × 10^3^ A/W, respectively. The photoresponse times were ~22.6 ms, 11.6 ms, and 9.7 ms. This behavior was due to the broadband light response of SnSe NSs and fast transportation of photocarriers between the monolayer graphene and SnSe NSs.

## 1. Introduction

Graphene-based photodetectors have attracted significant research interest [1,2,3]. However, the photo-responsivity is restricted by the low absorption of the one-atom thick graphene [4]. Several studies have reported that one of the effective ways is to construct hybrid structures, such as graphene-nanosheets (NSs) or graphene-quantum dots (QDs). This hybrid structure complements the low carrier mobility of nanomaterials and exhibits excellent chemical stability under incident radiation. On the other hand, nanomaterials can absorb incident light more efficiently than graphene [5]. A built-in electric field exists at the interface of this hybrid structure due to the energy offset, which can improve the dissociation efficiency of electrons and holes pairs as well as increasing the photoresponsivity [6]. Z. Sun et al. and Y. Sun et al. successively reported two kinds of photodetectors, which are based on oil phase QDs-graphene, these two devices can respond 895 nm light with a responsivity of 10^7^ AW^−1^ and 405 nm light with a responsivity of 10^3^ AW^−1^, and the sense time is about 0.26 s and 0.52 s, respectively [7,8]. Meanwhile, S. Lai et al. demonstrated a UV photodetector based on carbon nitride NSs -graphene, showing a responsivity of 10^3^ AW^−1^ and a photodetector based on graphene-graphene QDs was reported by M. Huang et al., reaching up to 4 × 10^7^ AW^−1^ under UV light, but the response time was 10 s [9,10].

The wet chemical methods, which are often used to synthesize uniformly dispersed NSs. Some organic solvents such as pyridine, oil amine and isopropanol are used intensely in this process [7,8,9,10]. Due to their high boiling point and surface tension [11], it is difficult to remove completely these solvents in subsequent experiments and the chemical groups may coat the surface of nanomaterials, leading to degradation of its charge transfer property. Moreover, the light wavelength of sensing in this hybrid graphene photodetectors is mainly based on the absorption band of NSs or QDs. Consequently, it turns out to be vital to explore facile and green methods to fabricate NSs or QDs and apply them with graphene to achieve broadband and sensitive detection.

In recent years, various efforts have been developed for fabricating 2D NSs, such as one-pot synthetic method [12], microwave hydrothermal technique [13], Li-intercalation exfoliation [14], ultrasonic liquid exfoliation [15,16], liquid metal synthesis [17,18]. SnSe, a member of layered IV–VI chalcogenides materials, is a promising candidate for photodetectors and solar cells due to the low cost, low toxicity, stability, a high coefficient of light absorption [19,20,21,22,23,24]. Zhao et al. utilized a vapor phase method to synthesis single-crystal SnSe NSs and demonstrated the field-effect transistor (FET) based on SnSe NSs. Under the white-light illumination, the device showed a photoresponsivity of ~330 A·W^−1^ at a bias voltage of 0.1 V [19]. Compared to the bulk SnSe, there are adjustable band gap, larger specific surface areas and more exposed active sites in the SnSe NSs [25].

Here, we prepared few-layer SnSe NSs in water-ethanol (0.7/0.3 ratio) using a facile ultrasonic liquid exfoliation, followed by the systemic investigation of the morphology and microstructure of the SnSe NSs. In addition, we used femtosecond transient optical absorption spectroscopy to study the transition process of photo-excited carriers in the SnSe NSs. Three decay components are resolved in the visible wavelength regime with a decay time of τ_1_ = 0.77 ps, τ_2_ = 8.3 ps, and τ_3_ = 316.5 ps, respectively. As a demonstration, we fabricated a SnSe NSs–graphene hybrid phototransistor using the drop-casting method. Given the wide distribution of SnSe NSs in the graphene channel, the phototransistor, with a wide channel length of 200 µM, displayed a broadband and sensitive detection ability of light radiation from 405 to 785 nm. The photo-responsivity was up to1.75 × 10^4^ A/W, 4.63 × 10^3^ A/W, and 1.52 × 10^3^ A/W, respectively for the 405, 532, and 785 nm light. The fast photo-response time was ~22.6 ms, 11.6 ms, and 9.7 ms, respectively.

## 2. Materials and Methods

### 2.1. SnSe NSs Preparation

SnSe bulk (99.9%) was purchased from Shenzhen Six Carbon Technology. In this study, the SnSe NSs was prepared using the liquid phase-exfoliated (LPE) method as follows: 50mg SnSe bulk was first ground to form SnSe plates which were added into 50 mL water-ethanol (0.7/0.3 ratio), an eco-friendly solvent, to form the SnSe suspension. The suspension was then treated by bath sonication for 24 h with a sustained energy power of 200 W, followed by a probe sonication process for 6 h with a sustained energy power of 500 W. The on/off cycle times were 4 s/6 s. It is worth noting that the temperature was maintained below 30 °C in the probe sonication process using an ice bath. Finally, the above dispersion was centrifuged at 3000 rpm for 30 min, followed by collection of the supernatant which was named as SnSe NSs.

### 2.2. Characterization

The morphology of the SnSe NSs was measured using high resolution transmission electron microscopy (HR-TEM, Tecnai G2 F30), atomic force microscope (AFM, Dimension Edge, Bruker, Middlesex County, MA, USA), and scanning electron microscope (SEM, Phenom Pro, Amsterdam, The Netherlands). The Raman spectroscopy was measured under a 514.5 nm laser (Horiba Labram HR Evolution). The optical absorption performance was measured using a UV-vis spectrum (Shimadzu, UV-1700) with a range of 250–1000 nm. X-ray photoelectron spectroscopy (XPS) was recorded using Al-Ka radiation PHI Versa ProbeII.

### 2.3. Pump-Probe Test

A regenerative amplifier laser system (Coherent, Legend Elite) with mode-locked Ti-sapphire, which generates 35-fs pulse at a repetition rate of 1 KHz and has a central wavelength of 800 nm, was used to measure the transient absorption spectrum. Most of the 800 nm output beam was used for pumping an optical parametric amplifier (Coherent, OperA Solo) in order to produce a pump beam. The pump beam was then chopped at 500 Hz. In addition, a little portion of the 800 nm output beam was fed to a sapphire crystal, located in the transient absorption spectrometer (Ultrafast Systems, HELIOS Fire), for the purpose of generating a broadband white light supercontinuum. An array detector and a lock-in amplifier were used to detect the intensity of the probe pulse, particularly pump beam off (I_o_) and pump beam on (I_ex_), after introducing the sample. A 400 nm (3.1 eV) pump pulse was then used to obtain transient absorption spectra. The absorption data was recorded as ΔA = logI_o_/I_ex_, and the time resolution was ~100 fs. All the measurements were performed under ambient temperature.

### 2.4. Measurements and Preparation of the Photodetector

The thickness of the SiO_2_ layer was approximately 300 nm. Source and drain electrodes (Cr:Au = 10 nm:90 nm) were formed on the substrate through thermal evaporation, and the channel length (L) and channel width (W) were defined as 0.2 and 2 mm, respectively, using shadow mask. A high quality monolayer graphene was transferred onto SiO_2_/Si substrates using the method reported by Liu et al. [26]. A micropipette was used to introduce 5 μL of SnSe NSs onto the graphene film in the channel and the device was then baked for 30 min in the glove-box. The temperature in the glove-box was set at 60 °C in order to remove the solvent and improve the interface contact between NSs and graphene. The process was repeated three times due to the stochasticity in the drop-casting process, followed by checking of the channel using SEM. A Keysight 4200 semiconductor parameter was used for measuring the photocurrent, and three substantive LED lasers (wavelength: 405, 532, and 785 nm) were purchased from Ocean Optics. A 250 W Xenon lamp coupled with the 300,150 monochromator was used as the light source to measure optical response of the device.

## 3. Results

SnSe NSs dispersed in water-ethanol were synthesized from SnSe bulk using four main processes of grinding, water bath and ice-bath probe sonication, and centrifugation as shown in Figure 1 (see the SnSe NSs preparation). The weak van der Waals forces were broken and the surface tension of solvent-NSs was matched to the attractive forces of NSs-NSs, thereby leading to the formation of a stable dispersion. Figure 1 also shows the crystal structure of SnSe bulk.

Figure 2 illustrates typical characterization of the as-prepared SnSe NSs. The morphology of the SnSe NSs was characterized by TEM and AFM. Figure 2a shows the TEM image of SnSe NSs, which indicates NSs with non-uniform sizes. The inset high resolution TEM image in Figure 2a demonstrates the distribution of lattice spacing stripes of SnSe NSs and the lattice spacing was about 0.301 nm. There is approximately 92° of intersection angle at the lattice fringes, which corresponds to the (011) planes of the orthorhombic structure of SnSe crystal [13]. Figure 2b shows the AFM images of the SnSe NSs on the substrate, which reveals that the SnSe NSs have the non-uniform size and different thickness. The typical thickness was found to be ~12 nm and 8 nm (Figure 2b inset) by a random line, respectively. The thickness of monolayer SnSe is ~1 nm, but it is difficult to get monolayer NSs due to the strong inter-layer interactions by the lone-pair electrons of Se [27], the probed SnSe NSs have a layer number around 10 but this thickness cannot cover all SnSe NSs in the solution [12]. Therefore, the as-prepared SnSe NSs have widely dispersed electronic band gaps. In addition, Figure 2c gives a closer image of a single SnSe NSs image and we can see that the uniform distribution of Sn and Se elements of SnSe NSs. The UV-vis absorbance spectroscopy was measured to investigate the optical properties of SnSe NSs (Figure 2d). It is clear that SnSe NSs shows a high absorption from 300 nm to 800 nm. The indirect bandgap of SnSe NSs is calculated based on Tauc’s equation (αhν)^1/2^ = A(hν − E_g_). By substituting the data of UV-vis-NIR spectra, the tauc plot can be obtained in the inset of Figure 2d. The band gap value is 1.34 eV, which is in agreement with the reported results [15,28]. In an indirect band gap semiconductor, indirect electron transitions require electrons interact not only with the photons, but also with lattice vibrations in order to either gain or lose phonons. Thus, indirect band gap semiconductors show broad absorption in the measuring range. This phenomenon can be observed in other layered IV–VI chalcogenides NSs [16,29].

Raman spectroscopy was employed to characterize bulk SnSe and SnSe NSs (Figure 3a). According to the theoretical crystal symmetry, a total of 12 active Raman modes exist in SnSe bulk [30]. However, only three vibrational modes $B_{3g}^{1},$
$A_{g,}^{2},$ and $A_{g}^{3}$ located at 108 cm^−1^, 128 cm^−1^, and 149 cm^−1^ can be readily detected. The intensity of the Raman peaks is significantly reduced in the SnSe NSs. In addition, there was a slight red shift of the $B_{3g}^{1}$ mode between SnSe NSs and SnSe bulk, which can be attributed to the fact that $B_{3g}^{1}$ mode is an out-of-plane mode, and thus it is more sensitive to the layer numbers [31]. Reduction of the peak intensity and shift of the peak position are common in 2D materials such as graphene and phosphorene, which might be attributed to the decrease of the atomic thickness and structural defects [16]. The XRD pattern of bulk SnSe is shown in Figure 3b. All the peaks were consistent with peaks obtained in other studies and a strong peak (400) was observed at 2θ = 31.2° [13,14]. Furthermore, all the main diffractions of SnSe NSs (Figure 3b inset) were in accordance with the bulk above and there were no observations of SnO_x = 1,2_ peaks. The intensity of several peaks of SnSe NSs decreased or disappeared [e.g., (400), (201), or (002)], which might result from the defects on the exposed surface after the exfoliation process [32]. In order to determine the element composition of SnSe NSs, it was further characterized via the high-resolution XPS spectra analysis, as shown in Figure 3c,d. The ratio of Sn: Se elements was close to 1:1 and the two peaks, located at 494.9 eV and 486.6 eV, were in agreement with the Sn 3d_3/2_ and Sn 3d_5/2_ binding energies of SnSe, and the peaks located at 55.7 eV and 54.7 eV corresponded to the doublets of Se 3d_3/2_ and Se 3d_5/2_ [33].

Developing a complete research of the ultrafast charge carrier dynamics in SnSe NSs is vital for the operation and optimization of SnSe NSs-based optoelectronic applications. This study utilized ultrafast transient absorption (TA) spectroscopy, with measurement time resolutions of 100 fs, as a robust tool for tracking the real time charge carrier dynamics in SnSe NSs. In the TA setup, the pump pulse excited the electrons, which enhanced their transport from the valence band into the conduction band. The different transmission date was recorded using a white light continuum, while the differential absorption was recorded as ΔA = log I_o_/I_ex_. The I_o_ and I_ex_ values represent the intensity of the probe pulse after introducing the sample under the conditions of pump pulse off and pump pulse on, respectively. ∆A was used to describe the absorption capability, which is a unit of the optical density (OD).

Figure 4a shows the TA map as the function of probe wavelength from 500 to 700 nm and delay time. The peak fluence of the pump pulse was 10 μJ/cm^2^ and the results indicate a broadband excited-state absorption (ESA) feature, where the absorption is enhanced because of the intra-band transitions. The effective mass of the hole is much larger than that of the electron, which means that the ESA is mostly due to the electron transition [34]. The electrons within the conduction band are transferred to higher electronic states after absorbing photons. This intra-band transition process is assisted by phonon to ensure momentum conservation [35]. The evolution of the spectra for SnSe NSs in the range of 1 to 100 ps is shown in Figure 4b, where one can clearly see that the broadband positively values the ESA feature. This indicates that the SnSe NSs have a broadband optical response region.

The ESA reflects the evolution of energized electrons. The TA signal at positive maxima around 640 nm (Figure 4c) was well fitted using tri-exponential model. It is a signal-wavelength kinetic analysis, and we obtain three time-constants: τ_1_ (0.759 ps), τ_2_ (8.01 ps), and τ_3_ (329 ps). In addition, the recovery time was very fast, which exactly demonstrates that SnSe NSs have a great potential for application in high-speed photodetectors. Globe fitting is an effective analytical method for complex spectra. We perform the globe fitting analysis of the TA spectra in the range of 500–700 nm. The kinetic curves of three principal components are shown in Figure 4d and we get three time-constants, τ_1′_ (0.77 ps), τ_2′_ (8.3 ps), and τ_3′_ (316.5 ps), which coincide with the fitting results of Figure 4c. The time constant τ_1_ (0.77 ps) can be attributed to the process of ESA, where the electrons were transferred to higher electronic states after absorbing photons. A strong quantum confinement effect and a reduced dielectric screening effect can result in strong Coulomb interactions between the electrons and holes, which induce the Auger processes effective for carrier capture by defects. The carrier capture processes via Auger recombination can realize even at relatively low carrier densities. There are different types of defect states in the SnSe NSs and defect states have different capturing rates for charge carriers. The other two time-constants τ_2_ (8.3 ps) and τ_3_ (316.5 ps) correspond to the two decay lifetimes of the carriers captured by the two different deep mid-gap defects states in SnSe NSs. One of the processes is fast, and the other process is slow. The defects are named as the fast defects and the slow defects [36,37,38]. In these processes, the excess energy of the electrons and holes is released by exciting other electrons or holes, thereby helping them attain higher energy states [39]. However, there is a different mode where the long recovery time results from the bonding excitons in the defect states, which block the fast recombination [16,40]. Further studies on the detailed recovery mechanism should be conducted.

We proposed the ultrafast dynamic processes of charge carriers in SnSe NSs based on recent reports [36] and results obtained in this study (Figure 4e). The electrons are first transported from the valence band into the conduction band under pump excitation. Afterwards, they are transported to the higher states after absorbing photons, and the electrons are assisted by phonon to satisfy the momentum conservation. Subsequently, the fast defects states will capture a part of the electrons and holes, within a few ps, through the Auger process. After occupation of the fast defect states, the remaining charge carriers will be captured by slow defects, thereby resulting in the slow recombination process.

The results of optical absorption (Figure 2d) and carrier dynamics (Figure 4) indicate the broadband absorption and fast carrier recovery time of SnSe NSs, which can be applied in a high-speed and broadband photodetector. A phototransistor with SnSe NSs–monolayer graphene was constructed, three different lasers (405, 532, 785 nm) were employed to evaluate its optoelectronic properties and all the photoelectric measurements were conducted in the glove box. An n-doped silicon substrate was cleaned using ultrasonic machine in order to fabricate the device, and then processed using O_2_ plasma. The electrodes were formed by shadow masking onto the substrate. On the other hand, the monolayer graphene-field devices were prepared using the wet-transfer method, followed by drop-casting of SnSe NSs on the graphene channel. Figure 5a shows the optical response of the phototransistor from 400 to 1000 nm measured under V_DS_ = 5 V. The spectral response of the device basically follows the absorption spectrum of the SnSe NSs. The schematic of a typical SnSe NSs–monolayer graphene hybrid photodetector is shown in Figure 5a inset. Figure 5b shows SEM image of the SnSe NSs on the top of graphene channel. The image reveals a discrete distribution of SnSe NSs on graphene, which leads to the formation of junctions between the SnSe NSs and graphene. In this study, the photocurrent (I_ph_) was calculated according to the expression of I_ph_ = I_light_ − I_dark_. Figure 5c shows the typical photocurrent curves as a function of the source–drain bias (V_DS_) with different excitation wavelengths. There was no photocurrent change of the monolayer graphene device under the light. It is clear that the photocurrent increases linearly with the rise of V_DS_ when the SnSe NSs was decorated on the graphene film. The highest photocurrent, a large value of ~3.00 mA, was obtained under 405 nm light illumination (V_DS_ = 5 V). In addition, a photocurrent of ~2.04 mA was observed under light illumination of 532 nm. Increasing the wavelength leds to the hybrid device displaying a photocurrent of ~0.95 mA under 785 nm illumination. The current change of photodetector can be attributed to the carrier accumulation in graphene, which are formed from charge transfer after optical excitation. The photoresponsivity (R_ph_) is calculated as R_ph_ = I_ph_/P, where P is defined as incident optical power. Indeed, it is found that the maximum responsivity is about 1.75 × 10^4^ A/W while illuminated at 405 nm with a radiation density of 23.6 μW/cm^2^, as shown in Figure 5d. It is worth noting that the channel length (200 μM) was larger than the length reported in previous SnSe-based photodetector studies [12,13,19,40]. Figure 5d also demonstrates that the responsivity decreases with the increase of light wavelength. The maximum responsivity under the 785 nm is found to be 1.52 × 10^3^ A/W at the power density of 155.2 μW/cm^2^.

The responsivity as a function of radiation density for different wavelength is summarized in Figure 6a. It can be seen that the corresponding responsivity decreased nonlinearly when the incident power was turned up. This may be due to the increase of the concentration of photoexcited carriers under higher incident power, which results in the carriers forming an electrical field and the formed electrical field is in reverse to the built-in field, thereby hindering the transport of photogenerated electron-hole pairs. The same change was reported in previous studies which constructed monolayer graphene photodetectors with a similar structure [5,6,7,8,9,10,41,42]. Furthermore, the responsivity reduced as the wavelength of light increases, which basically follows the absorption spectrum of the SnSe NSs (Figure 2d). The dependence between the incident light power density and the photocurrent can be determined using the following law: I_ph_ = *C*P*^a^*, where *C* represents the proportional constant and the index, and *a* denotes the ideal factor of photocurrent to light power density. For the ideal photodetector, the *a* = 1.0. The *a* value can be determined by fitting the curves as shown in Figure 6b and the ideal factor *a* are displayed in Table 1. The values of *a* are vary from 0.32 to 0.50. which demonstrate that the carrier generations exist in the graphene channel [43]. It is clear that the photocurrent gained in photodetectors consisting of SnSe NSs and monolayer graphene is much higher than the performance of the photodetectors made from graphene only. This can be attributed to the large and broad absorption of SnSe NSs and the charge transfer between SnSe NSs and graphene. However, the high responsivity is restricted to the bandgap absorption of SnSe NSs and this is a common characteristic for photodetectors having a similar structure, as reported in previous studies [7,8,9,10,41,42].

Next, we characterized the typical transfer curves (V_DS_ = 0.5 V) of all the devices, before and after SnSe NSs decoration, under varied illumination of wavelength, as shown in Figure 6c. After the decoration of SnSe NSs, the Dirac point of the hybrid device was located at +8 V position and had a higher conductivity under dark condition when compared with the only monolayer graphene device. There was an obvious p-doping effect in the channel and this is reasonable because SnSe is a p-type semiconductor, and the Fermi level is lower than graphene [13]. In addition, the Dirac point shifted to the more positive gate voltage under illumination. The maximum shift of the Dirac point was located at +18 V under 405 nm light and the minimal shift of the Dirac point (under 785 nm) may be due to the low excited photon energy or low optical absorption. Probably, the interface contact between SnSe NSs and monolayer graphene facilitates the injection of photogenerated holes from SnSe NSs into monolayer graphene upon illumination, whereas photoproduced electron are trapped in the SnSe NSs. This process effectively suppresses he recombination of photogenerated carriers and increases the number of holes in graphene, which in turn gives a clear photocurrent [7,43,44,45]. Figure 6d displays the schematic of photo-response scenario, which can further explain this phenomenon.

From the results of carrier dynamics of SnSe NSs (Figure 4), the hybrid device could have a high optical response speed. A series of channel photocurrent responses were performed at different wavelengths to demonstrate the sensitivity of the photodetector. The incident power density in all the responses was 155.2 μW/cm^2^. Figure 7a shows response of the photodetector to 405 nm light and the photocurrent can effectively rise and fall. A steady channel photocurrent of about 25.0 μA at V_DS_ = 0.05 V was observed when the light was on. The rise time and fall time were found at 22.6 and 24.6 ms, respectively (Figure 7b). Moreover, the stable channel photocurrent was about 18.5 μA at V_DS_ = 0.05 V under 532 nm light (Figure 7c). The rise time became more sensitive despite there being a reduction of channel photocurrent, which were determined to be 11.6 and 12.6 ms, respectively (Figure 7d). In addition, the photodetector still exhibited a clear channel photocurrent of ~8.0 μA at V_DS_ = 0.05 V under 785 nm light (Figure 7e). The rise and fall times of 9.7 and 16.6 ms, respectively (Figure 7f), suggest a faster photoresponse time than previous photodetectors based on hybrid graphene [5,6,7,8,9,10,41,42]. It was clear that there were two processes in the channel current, which were the fast process and the slow process. The fast increase after turning on the light can be attributed to the fast transfer process of carriers from SnSe NSs to graphene, while the electron-hole recombination may be deferred when the light is off due to the spatial separation of electrons and holes leading to a slow decrease of channel current [10].

## 4. Conclusions

In summary, ultrathin SnSe NSs were successfully synthesized using a facile and environment-friendly LPE method. Exfoliation of SnSe into few-layered NSs was confirmed by TEM, AFM, UV−vis absorption, Raman, XPS and XRD measurements. The carrier dynamics of SnSe NSs was studied by pump–probe test and the results exhibited that three decay components were resolved in the visible wavelength regime: τ_1_ = 0.77 ps, can be attributed to the process of ESA and two decay times of τ_2_ = 8.3 ps, and τ_3_ = 316.5 ps were ascribed to the processes of charge carrier being captured by the fast and slow defects, respectively, which provided fundamental guidance on the ultrafast optical and optoelectronic properties of this novel material. Application of the SnSe NSs for an absorption medium on the monolayer graphene resulted in the photodetector exhibiting high responsivity and a fast response speed from the 405 nm to 785 nm. The obtained results indicated that the responsivity was up to 1.75 × 10^4^ A/W, 4.63 × 10^3^ A/W, and 1.52 × 10^3^ A/W under 405, 532, and 785 nm light, respectively, and the photoresponse time was ~22.6, 11.6, and 9.7 ms. The photosensing mechanisms arise from several important factors, such as the high and broad absorptivity of SnSe NSs, high conductivity of monolayer graphene, the charge transfer between SnSe NSs and graphene and the fast carrier recovery time in the SnSe NSs. In addition, this study has also systematically revealed the effects of light wavelength and light power density on the behavior of photoresponse. The advantages of environment-friendliness and a facile fabrication process in this study indicate that the SnSe NSs could be used as a new 2D semiconductor nanomaterial. The broad photodetection performance demonstrates the obtained SnSe NSs-based monolayer graphene photodetector can be an excellent candidate for photonics device.

## Figures and Tables

**Figure 1 nanomaterials-11-00049-f001:**
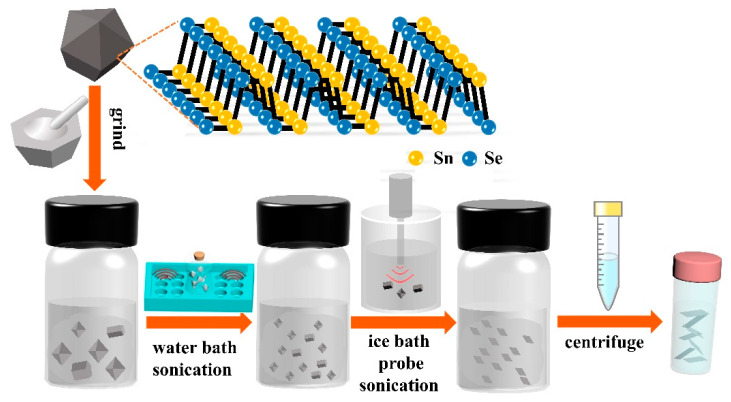
Schematic illustration of LPE to form SnSe NSs.

**Figure 2 nanomaterials-11-00049-f002:**
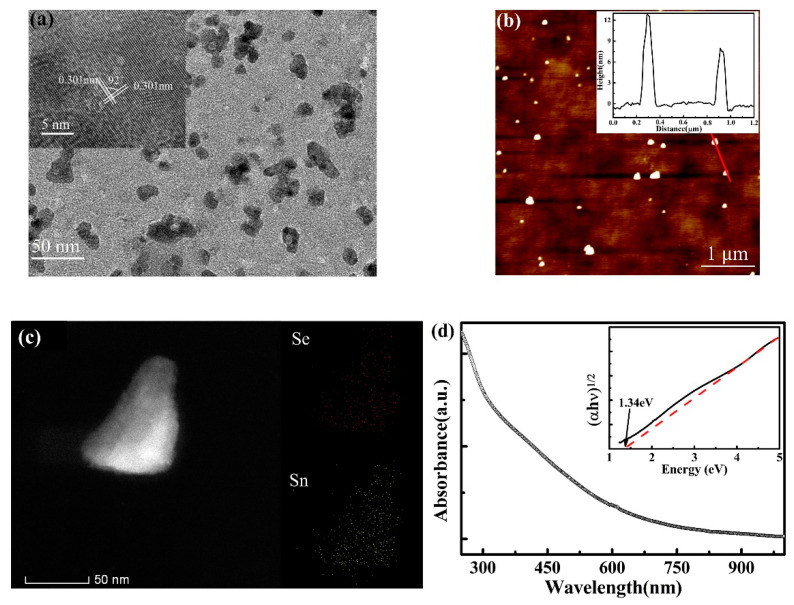
(**a**) TEM image of SnSe NSs, the inset shows HR-TEM image of SnSe NSs (**b**) AFM image of SnSe NSs, inset: the thickness histograms along the red line. (**c**) A single few-layer SnSe NSs and elemental mapping of Sn and Se. (**d**) UV−vis absorption spectrum of the SnSe NSs dispersed in water-ethanol solvent, inset: indirect bandgaps were determined from plots of (αhν)^1/2^ versus phonon energy.

**Figure 3 nanomaterials-11-00049-f003:**
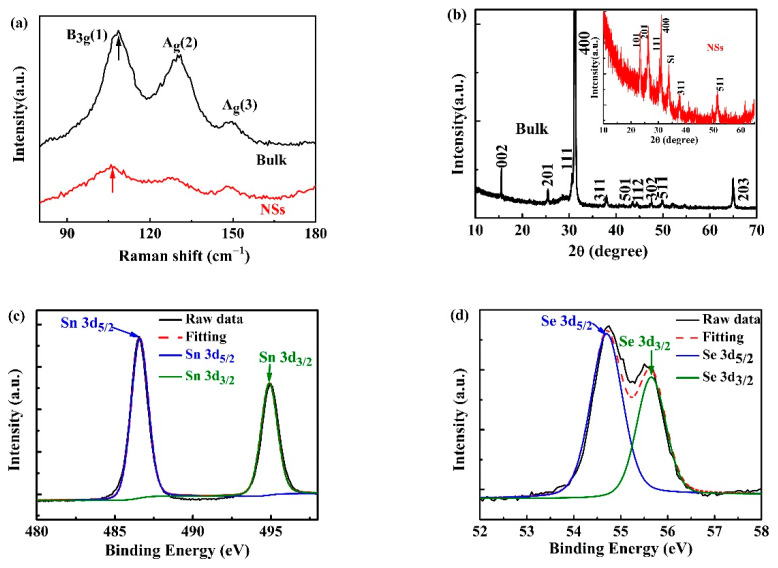
(**a**) Raman spectra and (**b**) XRD patterns of the bulk SnSe and SnSe NSs. High-resolution (**c**) Sn 3d and (**d**) Se 3d region XPS spectra of SnSe NSs.

**Figure 4 nanomaterials-11-00049-f004:**
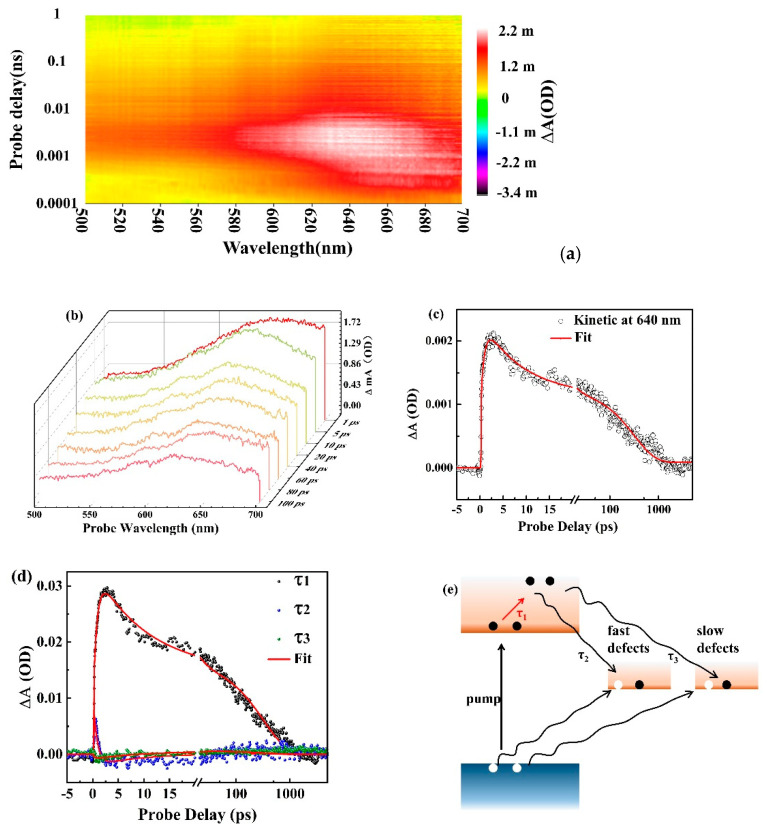
(**a**) The transient absorption map of SnSe NSs as the function of both delay time and probe wavelength. (**b**) The transient absorption spectra of SnSe NSs with a 400 nm pump pulse in the range of 1–100 ps. (**c**) The transient absorption signal measured from SnSe NSs with 640 nm probe pulse. (**d**) The globe fitting analysis of the TA spectra in the range of 500–700 nm. (**e**) Schematic illustration of carrier dynamics in SnSe NSs. The black curved arrows indicate the charge carrier be captured by the defects via Auger process and the red arrow indicate the ESA.

**Figure 5 nanomaterials-11-00049-f005:**
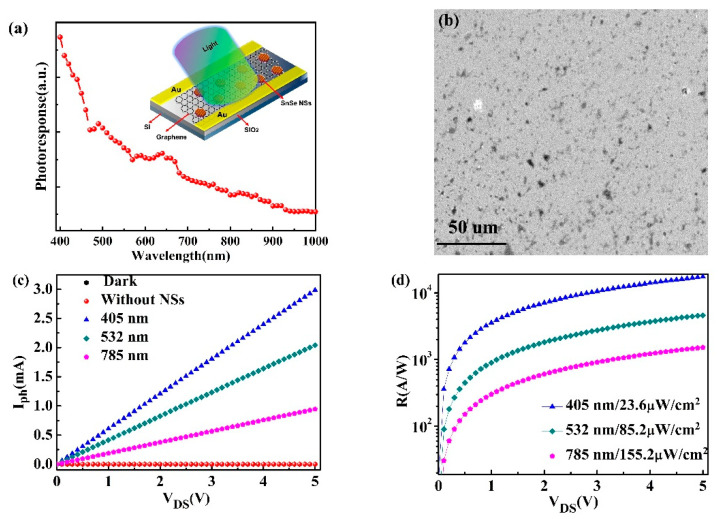
(**a**) Spectral response of the graphene–SnSe NSs photodetector. Inset: schematic diagram of the graphene–SnSe NSs photodetector. (**b**) SEM image of the SnSe NSs on the graphene film. (**c**) Photocurrent of the hybrid device under different light wavelength as a function of V_DS_ (Light power density of 155.2 μW/cm^2^, V_G_ = 0 V). The photocurrent for the only graphene device shows a negligible effect. (**d**) Responsivities as a function of V_DS_ (V_G_ = 0 V) under different light wavelength, each with the lowest power intensity.

**Figure 6 nanomaterials-11-00049-f006:**
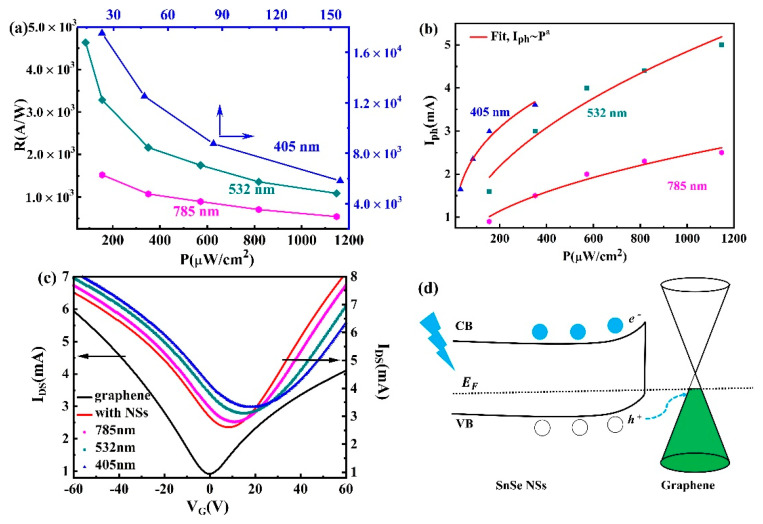
(**a**) Responsivity as a function of incident power at 405, 532, and 785 nm, respectively, and (**b**) the light power density dependence of the photocurrent under 405, 532, and 785 nm, respectively (V_DS_ = 5 V, V_G_ = 0). (**c**) Transfer characteristics of the graphene–SnSe NSs photodetector under 405, 532, and 785 nm, respectively. (Light power density of 155.2 μW/cm^2^, V_DS_ = 0.5 V). (**d**) The illustration of the photo-response mechanism.

**Figure 7 nanomaterials-11-00049-f007:**
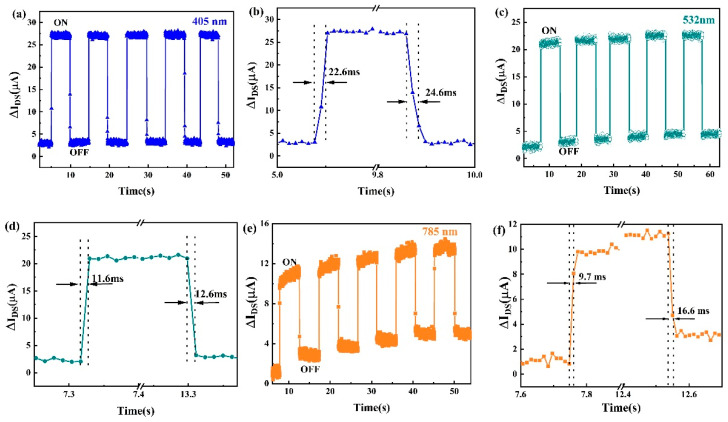
Channel photocurrent response to on/off light illumination (**a**) 405 nm (**c**) 532 nm (**e**) 785 nm for several cycles and time-resolved channel photocurrents to show the rise and fall times for (**b**) 405 nm (**d**) 532 nm (**f**) 785 nm. (Light power density of 155.2 μW/cm^2^, V_G_ = 0 V, and V_DS_ = 0.05 V).

**Table 1 nanomaterials-11-00049-t001:** Indexes *a* obtained by fitting I_ph_~P.

Wavelength	405 nm	532 nm	785 nm
***a***	0.32	0.50	0.47

## Data Availability

The data are not publicly available due to privacy.

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
