# Peer review of "SnSe Nanosheets: From Facile Synthesis to Applications in Broadband Photodetections"

_nanomaterials, 2020, doi:10.3390/nano11010049_

Round 1
Reviewer 1 Report
The manuscript by X. Li et al. demonstrate the optical and optoelectrical properties of SnSe and SnSe/graphene heterostructure. The carrier dynamics of SnSe were investigated using the pump probe test, and the optoelectrical properties of the SnSe/graphene photodetector were well characterized. The manuscript is well written in as scholarly manner. Therefore, I recommend this manuscript to be published in nanomaterials after addressing the following issues.
- It is recommended to provide an optical image of SnSe NSs on a 300nm SiO2/Si substrate
- Electrical properties of SnSe are missing. Authors are recommended to show the electrical properties of the SnSe-based FET.
- UV-vis in Figure 2d does now show the absorption peaks that correspond to the band gap of SnSe. Why is this so?
- Figure 4c and d are repetitive.
- What are the slow and fast defects. Please specify.
- In line 215 of page 7, “Fig 1d” should be “Fig 2d” and “Fig 2” should be “Fig 4”.
- How are the carrier dynamics of SnSe NSs in Figure 3 affect the optoelectrical properties of the SnSe/graphene photodetector?
Reviewer 2 Report
The authors synthesized 2D SnSe by sonication in water-ethanol solvent. They investigated the carrier property in 2D SnSe using femtosecond transient absorption spectroscopy. They obtained this in visible wavelength range. They fabricated photodetectors, which integrated 2D SnSe and graphene. It is a good paper and merit publication I only have a few somments as below 1- The authors could investigate the possibility of producing SnSe using liquid metal synthesis process, similar to what presented in Nature Communications volume, 11, 3449 (2020) and Advanced Materials, Volume32, Issue45, November 12, 2020, 2004247 2- Mention that the exfoliation of SnSe is challenging due to the strong inter-layer interactions by the lone-pair electrons of Se and reference Nanoscale 10, 22474–22483 (2018) 3- Measure and present the optical responsivity of the device 4- The representation in fig 6 d is not correct and should be amended 5- Expand the conclusion Altogether it is a very good paperAuthor Response
Please see the attachment.

Round 2
Reviewer 1 Report
I have no further comments